# SynMob: Creating High-Fidelity Synthetic GPS Trajectory Dataset for Urban Mobility Analysis

**Yuanshao Zhu**[1,2]*, **Yongchao Ye**[1]*, **Ying Wu**[1,3], **Xiangyu Zhao**[2]†, **James J.Q. Yu**[4]†

[1] Southern University of Science and Technology
[2] City University of Hong Kong
[3] University of Leeds
[4] University of York
{zhuys2019, 12032868, 12059004}@mail.sustech.edu.cn
xianzhao@cityu.edu.hk
james.yu@york.ac.uk

## Abstract

Urban mobility analysis has been extensively studied in the past decade using a vast amount of GPS trajectory data, which reveals hidden patterns in movement and human activity within urban landscapes. Despite its significant value, the availability of such datasets often faces limitations due to privacy concerns, proprietary barriers, and quality inconsistencies. To address these challenges, this paper presents a synthetic trajectory dataset with high fidelity, offering a general solution to these data accessibility issues. Specifically, the proposed dataset adopts a diffusion model as its synthesizer, with the primary aim of accurately emulating the spatial-temporal behavior of the original trajectory data. These synthesized data can retain the geo-distribution and statistical properties characteristic of real-world datasets. Through rigorous analysis and case studies, we validate the high similarity and utility between the proposed synthetic trajectory dataset and real-world counterparts. Such validation underscores the practicality of synthetic datasets for urban mobility analysis and advocates for its wider acceptance within the research community. Finally, we publicly release the trajectory synthesizer and datasets, aiming to enhance the quality and availability of synthetic trajectory datasets and encourage continued contributions to this rapidly evolving field. The dataset is released for public online availability https://github.com/Applied-Machine-Learning-Lab/SynMob.

## 1 Introduction

Urban mobility analysis is an essential aspect of modern urban planning and development [45]. It provides ongoing insight into the movement patterns of individuals and populations within urban environments, informing decisions related to infrastructure design, transportation planning, and policy decisions [46, 48]. Central to the progress of this analysis is the trajectory data, which captures the movement of individuals temporally and spatially. This type of data is invaluable for understanding the social and environmental impacts of urban transportation, as it allows researchers and practitioners to model and predict movement patterns [28, 15], identify congestion points [12], and optimize transportation networks [24].

Despite the undeniable value of trajectory data in urban mobility analysis, several challenges have hindered its widespread utilization. Firstly, there is a noticeable lack of publicly available trajectory

---

*These authors contributed equally to this work

†Corresponding author

37th Conference on Neural Information Processing Systems (NeurIPS 2023) Track on Datasets and Benchmarks.

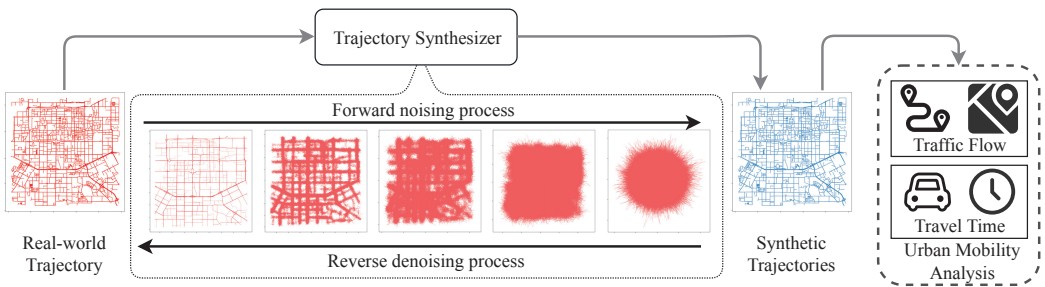

Figure 1: The illustration of using synthesize trajectory dataset for urban mobility analysis. Here, the trajectory synthesizer uses a diffusion model to learn complex spatial-temporal motion patterns, enabling the synthetic trajectory dataset with high-fidelity.

datasets. This scarcity is primarily due to the proprietary nature of such data, which are typically collected and owned by private entities or government agencies [32, 49]. Secondly, strict regulations and data privacy concerns further limit the accessibility of trajectory data. The sensitivity of this data can reveal personal information about personal activities and routines, necessitating strict privacy safeguards [19, 40]. Lastly, even when trajectory data is available, it often suffers from poor format and quality. Inconsistent data collection methods, missing data points, and noise can significantly degrade the utility of these datasets, making them less useful for rigorous, high-quality research in urban mobility analysis [23, 31]. As a result, the lack of publicly available, privacy-free, and high-quality trajectory data presents a significant obstacle to the progress of the urban computing community. Addressing the above challenges calls for creating alternative trajectory data sources that are accessible and available for research.

Figure 1 illustrates a promising solution that resorts to synthetic datasets: synthesizing trajectories by learning real trajectory distributions. Based on the synthesized trajectory, equivalent urban mobility analysis results can be expected while avoiding privacy leakage. However, due to the complex spatial-temporal properties inherent in trajectory, generating the proper dataset is a non-trivial task. Real-world trajectories are characterized by intricate spatial patterns and temporal dependencies, reflecting the variety and dynamics of human mobility [45, 20]. For example, different areas within a city (e.g., commercial, residential) present various functions and population densities, resulting in non-independent and identically trajectory distribution [50]. Moreover, the spatial aspect of trajectory data is often intertwined with the temporal aspect, creating a high-dimensional data structure that is challenging to model accurately [44, 18, 6]. For example, the same location can exhibit different mobility patterns at different times, and the same individual can exhibit different mobility patterns at different locations [15, 28]. In addition, individual location at a given time is influenced by their previous location, as well as by the time of day, traffic conditions, and various other contextual factors [19, 50, 43]. Therefore, successfully synthesizing trajectory datasets advocates a method that accurately captures and integrates these complex spatial-temporal behaviors.

To confront these intricacies, our objective centers on crafting a high-fidelity synthetic trajectory dataset tailored for urban mobility analysis, followed by a thorough evaluation of its practicality. Our research revolves around probing: 1) The feasibility of devising a trajectory generation technique that adeptly captures spatial-temporal characteristics for dataset synthesis. 2) The ability of the synthesized dataset to mirror the intrinsic geo-distribution and statistical attributes while retaining the original utility. To navigate these queries, our work offers the ensuing contributions:

- We introduce a groundbreaking high-fidelity synthetic trajectory dataset, birthed using the diffusion model. This dataset not only bridges the existing chasm in data accessibility and privacy but also propels the creation of synthetic trajectory data. By equipping community researchers with tools for generating premium trajectory datasets, we envision a surge in urban traffic research and its subsequent innovations.

- We embark on exhaustive analyses and empirical studies to authenticate our methodology. By juxtaposing the synthesized datasets with their real-world analogs across diverse aspects, we underscore their striking resemblance. Such validation accentuates the robustness and adaptability

of synthetic datasets in urban mobility studies, thereby fortifying the methodological foundation of this academic domain.

## 2 Background

Since our focus is on synthesizing GPS trajectory data for urban mobility analysis, a brief review of the related work on trajectory data in urban mobility analysis and the corresponding trajectory datasets is presented in this section.

**Trajectory data in urban mobility analysis:** The cornerstone of urban mobility analysis is the well-conceived utilization of trajectory data, which captures the spatial and temporal dimensions of individual movements within these spaces [48, 46]. Based on trajectory data, researchers can investigate movement patterns within urban areas, providing valuable insights to inform various decisions, from infrastructure development and transportation planning to decision-making and resource allocation. One important application of trajectory data is travel mode identification [49], enabling researchers to identify common modes of transportation and travel habits, as well as to detect anomalies and changes in travel behavior over time [32]. This information can be used to optimize transport networks, improve traffic management and enhance urban infrastructure planning. In addition, trajectory data serves for traffic flow forecasting and traffic condition diagnosis [24]. By analyzing historical trajectory data, it can help researchers identify trouble areas and trends in the urban network, enabling them to predict future congestion levels and take proactive measures to alleviate them. Finally, trajectory data can be used for travel time estimation [19, 11, 38]. Analyzing travel times for past trips allows researchers to predict travel times for future trips, which can help commuters plan their trips more efficiently and help transportation providers deliver a favorable user experience. Besides, trajectory data can also be applied to route planning [5], similarity analysis [9], and public health analysis [1], but all of its potential benefits on urban mobility analysis are rooted in the availability of sufficient trajectory data.

Table 1: Summaries of existing trajectory datasets.

| Dataset | GPS trajectory | Availability | Data quality | Privacy | # Trajectory |
|---|---|---|---|---|---|
| GeoLife[3] [47] | ✓ | ✓ | ✗ | ✗ | 17,621 |
| T-drive[4] [39] | ✓ | ✓ | ✗ | ✗ | 10,357 |
| Porto[5] [22] | ✓ | ✓ | ✗ | ✗ | 1.7 million |
| Foursquare[6] [37] | ✗ | ✓ | – | ✗ | 104,478 |
| NYC[7] | ✗ | ✓ | – | ✗ | 1.1 billion |
| Taxi-Shanghai[8] | ✓ | ✗ | ✗ | ✗ | 1.2 million |
| GAIA[9] | ✓ | ✗ | ✓ | ✗ | 3.1 million |
| Ours (Synthetic) | ✓ | ✓ | ✓ | ✓ | unrestricted (customize) |

**Available Trajectory Dataset:** Considering the tremendous value of trajectory data in urban mobility analysis, there are several trajectory datasets available for urban mobility analysis. As concluded in Table 1, these datasets vary in source, coverage, granularity, and quality, and each has unique strengths and limitations. Geolife data is one of the most prestigious trajectory datasets, which was collected over three years (April 2007 to August 2012) by 182 users [47]. The dataset is widely applied in many research areas such as travel mode identification [8], location recommendation [3], and traffic flow analysis [41]. However, the trajectory is recorded by heterogeneous GPS devices, involves multiple sampling rates, and is limited by coverage, as they can only capture the behavior of a tiny population. Additional trajectory data, such as Porto [22], T-drive [39], and Taxi-Shanghai are collected by GPS devices mounted on the taxi. These datasets provide broader coverage and capture

---

[3]https://www.microsoft.com/en-us/research/publication/geolife-gps-trajectory-dataset-user-guide

[4]https://www.microsoft.com/en-us/research/publication/t-drive-trajectory-data-sample

[5]https://www.kaggle.com/datasets/crailtap/taxi-trajectory

[6]https://sites.google.com/site/yangdingqi/home/foursquare-dataset

[7]https://www.nyc.gov/site/tlc/about/tlc-trip-record-data.page

[8]https://cse.hkust.edu.hk/scrg

[9]https://outreach.didichuxing.com

the activity of many individuals, but they normally provide low-sampling rate data. In addition, the GAIA Initiative at Didi-Chuxing provides a series of high-quality and extensive datasets [10]. However, these datasets are not publicly available due to privacy concerns and proprietary restrictions, which restrict their accessibility and usability. Other datasets, such as Foursquare [37] and NYC, provide a large amount of location-based social network data. Such datasets lack the necessary contextual information to fully understand the mobility pattern, limiting the scope of research that can be conducted using these datasets [45, 49, 50]. As a result, while a range of trajectory data is available, with their availability is often hampered by a variety of challenges, including authorization issues, privacy concerns, and data quality matters. These challenges highlight the aspiration for alternative sources of trajectory data, such as synthetic datasets, which can overcome these limitations and improve the scope and quality of urban traffic analysis.

## 3   Technical Design

Given that the primary focus of this paper is the synthesized trajectory dataset, our discussion regarding the technical design of the diffusion model-based trajectory synthesizer is intentionally succinct in this section. Comprehensive details about the underlying principles and design intricacies can be found in previously published works, specifically in references [14, 26, 27]. Figure 1 illustrates the foundational distinction of the trajectory synthesizer we propose. Notably, it is the innovative incorporation of a diffusion model into its design. Historically, diffusion models have been predominantly utilized in the realm of nonequilibrium thermodynamics, a fact evidenced by studies such as [26, 14]. However, our approach seeks to repurpose this model for a radically different domain. Translating the concepts into the domain of urban mobility analysis, one can draw parallels between the inherent components of the diffusion model and the urban setting. Here, the term "particles" can be aptly likened to individuals navigating the intricate urban landscape. Meanwhile, the "medium" in which these particles move is analogous to the very fabric of the urban environment – its streets, alleys, and open spaces. The adoption of the diffusion model in trajectory synthesis is not just an academic exercise but brings forth two significant advantages. Firstly, the mechanism by which the diffusion model operates synthesizes data in a granular, incremental manner, starting from mere random noise as indicated by [27]. This intricate process ensures the resulting trajectories are purged of any traces of sensitive personal data, thus fortifying the privacy measures. Secondly, the inherent versatility of the diffusion model facilitates the simulation of an expansive range of spatial-temporal behaviors. This ensures that the synthesized dataset is not a monolithic block but instead, mirrors the multifaceted, heterogeneous nature observed in genuine real-world mobility patterns. Such a rich and comprehensive representation is indispensable, as it affords researchers the latitude to delve into a plethora of scenarios and phenomena. Consequently, it amplifies the scope, generalizability, and relevance of the derived insights and findings. Specifically, the trajectory synthesizer uses the following design:

- **Main Processes:** Our approach leverages the standard diffusion model process, encompassing both forward noising process and reverse denoising process [4]. As shown in Figure 1, the forward noising process involves iteratively adding noise to the original trajectory, transforming it into a Gaussian noise over a series of diffusion steps. On the other hand, the reverse denoising process works in the opposite direction. Starting from the Gaussian noise, it progressively removes noise to finally generate a high-fidelity trajectory (For more details about these two processes, please refer to [14]). This mechanism ensures that the trajectory synthesizer generates trajectories based on the learned spatial-temporal distribution, rather than directly manipulating real-world data.

- **Architecture:** Our trajectory synthesizer architecture builds upon the widely recognized denoising neural network, UNet [25], which has been tailored to align with our specific requirements. The architecture is optimized to handle the unique structure of trajectory data, ensuring precise noise level predictions at each diffusion step.

- **Quality Ensuring:** We employ a classifier-free diffusion guidance method [13] to strike a balance between the quality and diversity of synthesized trajectories. By integrating advances from the literature, notably the non-Markov diffusion process method [27], we enhance computational efficiency and generate high-quality trajectories in a condensed number of steps.

- **Training:** The training for our trajectory synthesizer draws from two significant trajectory datasets from the GAIA project [10]. While these datasets contain invaluable urban mobility insights, their proprietary nature restricts broad access. Our synthesizer aims to tap into the potential of these

datasets by producing simulated trajectories that are devoid of sensitive information. This approach not only maintains data privacy but also negates risks associated with reverse engineering. Practical adjustments to the dataset, such as length normalization, are also implemented for optimal model training.

# 4 Synthetic Dataset Analysis

Using the well-trained trajectory synthesizer, we generate two datasets based on the original data, i.e., SYN-CHENGDU and analysis of SYN-XI'AN. In this section, we perform a detailed analysis of the SYN-CHENGDU datasets, including a basic description and an evaluation of their quantities (The analysis of SYN-XI'AN is conclude in the **Appendix** A). Specifically, this dataset accurately reflects the characteristics and trends of real-world counterpart, making it valuable for downstream tasks and applications. However, it's essential to note that our primary goal of this synthetic dataset is not just to reproduce the original data but to provide an alternative when the original dataset is not available. The slight discrepancies observed also highlight areas for potential improvement in our method, and we are continuously working to refine our approach to narrow this gap further. We hope that this dataset can be used as a benchmark for future research in the field of urban mobility analysis. Compared to the original dataset collected from physical devices such as GPS devices or mobile phones, the synthetic dataset has the following advantages:

- **Privacy free:** It provides privacy protection by generating trajectories that do not contain any personally identifiable information or sensitive data.

- **High fidelity:** The synthetic dataset has high fidelity, with similar statistical features as the original dataset, making it a representative sample of real-world data.

- **Public availability:** The synthetic dataset can be made publicly available, enabling researchers and practitioners to use it for various purposes without violating regulations.

- **Scalability:** The trajectory synthesizer can generate an arbitrary amount of synthetic trajectories, addressing the problem of insufficient data and making the synthetic dataset highly scalable.

- **Enhancing diversity:** While the synthetic dataset is derived from real data, the synthesis process introduces variability and diversity, ensuring that the dataset is not just a mere replication but offers varied patterns that can be valuable for different applications.

Table 2: Dataset description of SYN-CHENGDU

| Type | Description |
| --- | --- |
| Format | pickle / geoparquet |
| Size | 4.39 GB |
| Value type | float64 |
| Time frame | 5 min |
| Sample interval | 3 s |
| Spatial coverage | lat: $30.65° \sim 30.73°$
lng: $104.04° \sim 104.13°$ |

## 4.1 Dataset Description

As concluded in Table 2, the synthetic trajectory dataset SYN-CHENGDU, stored in pickle format with a float64 value type, is readily accessible and compatible with a variety of data analysis tools. The dataset contains one million records, each representing an individual trajectory, organized into two parts: attribute information and trip trajectory. Among them, the attribute information is used to record the `departure time` $t_d$`, trip distance (meters), trip time` $t_a$`(seconds),` and `sampling points` $n$. Specifically, the data is sampled at 3-second intervals (same sampling rate as the original trajectory dataset), and the departure time is divided into 288-time frames with a 5-minute duration. Thus, we can derive the timestamp for any point based on the departure time ($t_i = t_d + i \times 3$ s). For the trip distance, we use the Vincenty [29] formula to calculate the relative distances of two contiguous points and accumulate them. These attributes record the temporal and

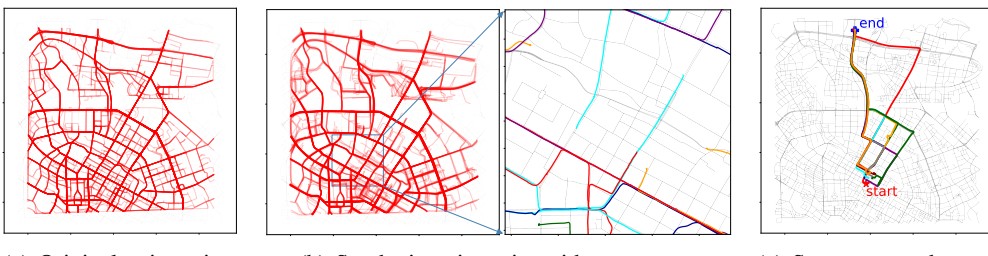

(a) Original trajectories.    (b) Synthetic trajectories with area zoom.    (c) Same start-end areas.

Figure 2: Comparative visualizations of original and synthetic trajectories. (a) and (b) server as a reference for the quality and accuracy of the synthetic trajectories. (c) depicts the diversity of synthetic trajectories based on the same start and end areas.

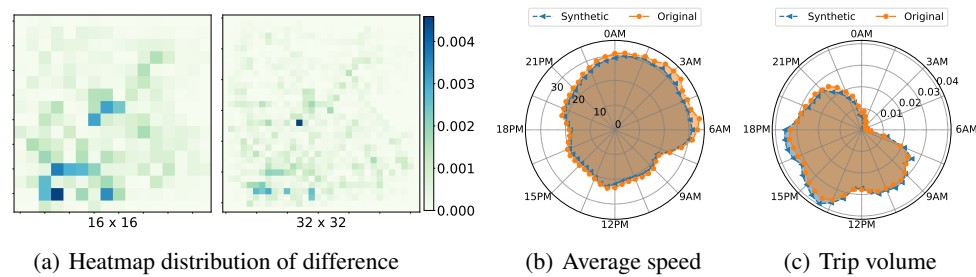

(a) Heatmap distribution of difference    (b) Average speed    (c) Trip volume

Figure 3: Spatial-temporal comparison of the synthetic dataset and original counterpart (SYN-CHENGDU). (a) Differences in heatmap of raw-synthetic (calculated by absolute) trajectories distribution. (b) and (c) The average speed and the number of trips throughout the day.

motion profiles of the trajectory and provide a wealth of information for analysis. In addition, the trip trajectory part consists of a series of consecutive data points represented by $\mathcal{P} = \{p_1, \ldots, p_n\}$, where $p_i = [\text{lat}_i, \text{lon}_i], i \in \{1, \ldots, n\}$ represents the latitude and longitude of the location. In terms of spatial coverage, the dataset spans a wide geographical area, encompassing various urban environments. This spatial coverage enables the analysis of spatially-dependent mobility patterns and phenomena.

## 4.2 Quality Evaluation

In this section, we provide a comprehensive quality evaluation of the synthetic dataset, focusing on its spatial-temporal distribution and its motion characteristics. For a better presentation of the quality of the synthetic dataset, we chose its counterpart in the real world as a reference comparison.

**Trajectory geo-distribution insight:** Figure 2 provides a comprehensive visualization of synthetic trajectories within the urban road network, reflecting several important characteristics of the synthesized data. In particular, Figure 2(a) and Figure 2(b) offer a side-by-side comparison of original and synthetic trajectories. Notably, the synthetic trajectories successfully adhere to the geo-distribution and sparse properties that are characteristic of the urban road network in the original counterparts. This observation suggests that the synthetic data generation process can effectively mimic real-world data, preserving its key spatial characteristics. This insight is further validated by Figure 2(b), portraying the consistency of synthetic trajectories with real-world valid routes and positional details. This feature suggests that synthetic trajectories hold road segment validity, which can support road-based spatial-temporal tasks, e.g., map matching. Furthermore, Figure 2(c) presents synthetic trajectories that maintain consistent start and end areas. This visualization emphasizes the synthesizer capable of generating trajectories exhibiting a wide range of behavioral profiles, further enhancing the diversity and complexity of the synthetic dataset.

**Spatial-temporal distribution:** Beyond trajectory visualization, this study offers an analysis of the spatial and temporal distribution embedded within the synthetic dataset, further substantiating its

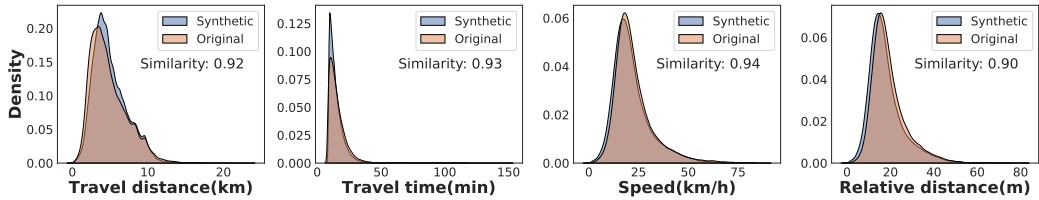

Figure 4: Comparative analysis of trajectory properties in original and synthetic datasets.

fidelity. Figure 3(a) presents a heatmap depicting the citywide distribution of trajectories, enabling a detailed comparison between the synthetic and original datasets. The results clearly show that the synthesized trajectory dataset can maintain a high degree of spatial distribution consistency, which provides confidence in urban traffic prediction based on the synthetic trajectory dataset. Figures 3(b) and 3(c) delve into the temporal dynamics of the synthetic dataset by capturing the variation in average speed and trip volume over 24 hours. Specifically, these figures highlight the ebb and flow of urban movement, showing a slower average speed and a higher number of trips during peak hours (9 AM – 6 PM). This pattern reverses during the early hours of the day (0 AM – 6 AM) when activity levels in the city typically diminish. The temporal pattern reflected in the synthetic dataset is strongly aligned with what would be expected based on real-world trends. This synchronization fidelity in temporal patterns is instrumental in facilitating reliable travel time estimations based on the synthetic dataset. In effect, the synthetic data is shown to accurately encapsulate the day-night cycle of urban mobility, further enhancing its utility and application potential.

**Trajectory properties:** Finally, we investigate the coherence at the trajectory property level. As depicted in Figure 4, we examine the similarities between the synthetic and original trajectories, specifically considering travel distance, travel time, speed, and relative distance. From our visualization results, it is evident that the synthetic dataset manifests a substantial concordance with the real-world data. To further quantify this consistency, we applied the Kolmogorov-Smirnov (K-S) test [21], a non-parametric method of comparing two one-dimensional distributions. The K-S statistics, a similarity measure, are reported as 0.92, 0.93, 0.94, and 0.90 for travel distance, travel time, speed, and relative distance, respectively. These results suggest that the distributions of these key attributes in the synthetic data closely match those in the original data, with the similarity exceeding 90% in all cases. Such high K-S statistics highlight the robustness of the synthetic data generation process, with the synthetic dataset showing a strong adherence to the trajectory-level properties observed in the original data. This conclusion underscores the utility of the synthetic dataset, its suitability for a range of research applications, and its potential to facilitate future studies in urban mobility.

## 5   Use Cases of Synthetic Dataset

To further demonstrate the practical utility and applicability of our synthetic dataset, we explore its use in two critical urban mobility tasks: traffic demand prediction and travel time estimation. These tasks hold significant importance in the field of urban mobility, with the former offering insights into future traffic conditions and the latter providing crucial information for trip planning. Through these applications, we aim to illustrate the versatility of our synthetic dataset and its capacity to synthesize reliable results in comparison with real-world data. We trust that the following analysis and case studies will validate the robustness of the synthetic dataset and provide further guidance for its use. All simulations are implemented in PyTorch and performed on computing servers with one NVIDIA RTX 2080Ti GPU.

### 5.1   Traffic Demand Prediction

**Problem definition:** Traffic Demand Prediction (TDP) is a cornerstone of urban mobility analysis, aiming to predict origin-destination demand in various areas within the city [42]. *Specifically, the purpose of TDP is to predict the vehicle inflow or outflow $a_d^t$ for a given area $d$ at time $t$.*

**Setup:** To predict traffic demand, we first preprocess the trajectory dataset (both original and synthetic) by grouping it into different time intervals and areas and then aggregating the traffic

Table 3: Data utility comparison by traffic demand prediction. Results are expressed as (original / synthetic / difference ratio).

| Methods | AGCRN | GWNet | DCRNN | MTGNN |
|---|---|---|---|---|
| RMSE | 6.91 / 6.50 / 5.93% | 6.90 / 6.53 / 5.36% | 7.29 / 6.48 / 11.11% | 6.81 / 6.41 / 5.87% |
| MAE | 4.64 / 4.43 / 4.53% | 4.65 / 4.47 / 3.87% | 4.88 / 4.45 / 8.81% | 4.58 / 4.39 / 4.15% |
| MAPE | 30.47/ 30.97 / 1.64% | 30.57 / 30.74 / 0.56% | 32.40 / 30.40 / 6.17% | 29.61 / 29.88 / 0.91% |

Table 4: Data utility comparison by travel time estimation. Results are expressed as (original / synthetic / difference ratio)

| Methods | TEMP | XGBoost | WDR | DeepTTE |
|---|---|---|---|---|
| RMSE | 290.32 / 282.33 / 2.75% | 271.56 / 256.19 / 5.66% | 258.64 / 247.29 / 4.39% | 216.93 / 193.34 / 10.87% |
| MAE | 182.74 / 174.42 / 4.55% | 175.20 / 167.75 / 4.25% | 149.81 / 140.05 / 6.51% | 132.95 / 121.31 / 8.76% |
| MAPE | 18.62 / 17.81 / 4.30% | 17.97 / 16.94 / 5.73% | 14.06 / 13.17 / 6.33% | 13.07 / 12.29 / 5.97% |

demand for each area and interval. In this case study, a time interval of 15 minutes is utilized, while the road network is partitioned by default into 144 areas, consisting of a 12×12 grids. To demonstrate the applicability of SYN-CHENGDU across a range of data-driven TDP tasks, we utilize four representative DNN-based models, namely, AGCRN [2], GWNet [35], DCRNN [17], and MTGNN [34], These models cover different variants of advance neural network, which have been widely used in the literature [16]. The performance metrics used for their comparison are Root Mean Squared Error (RMSE), Mean Absolute Error (MAE), and Mean Absolute Percentage Error (MAPE). For cross-validation, we partition the dataset (original / synthetic) into training, validation and test sets in a 7:1:2 ratio. The input and output steps of the model are both set to 12, i.e., we use traffic demand from the past 3 hours to forecast traffic demand for the next 3 hours. Further details and implementation specifics of these methods are provided in **Appendix** B.

**Result analysis:** Table 3 presents a comparison of the data utility in traffic demand prediction for both original and synthetic datasets using various neural network models. Each metric is reported for both original $\mathcal{M}_o$ and synthetic datasets $\mathcal{M}_s$, and the difference ratio $|\mathcal{M}_s - \mathcal{M}_o|$ (lower is better) is computed to assess the consistency performance. Overall, these models demonstrate similar performances when trained on both datasets, indicating the high fidelity of the synthetic dataset. Specifically, the difference ratio in performance metrics typically remains within a modest range (the maximum difference ratio is 11.11%), confirming that our synthetic dataset can effectively mirror the real-world scenario in traffic demand prediction tasks. It is worth noting that the DCRNN model exhibits a slightly higher difference ratio, particularly in RMSE and MAE. The minor divergence under this model serves as a reminder of the challenges faced when synthesizing data with intricate spatial-temporal dynamics. In conclusion, these results corroborate the suitability and practical utility of our synthetic dataset for traffic demand prediction, thereby reinforcing its potential for advancing research in urban mobility analysis.

## 5.2 Travel Time Estimation

**Problem definition:** Travel Time Estimation (TTE) is a pivotal aspect of urban mobility analysis. Its primary objective is to estimate the travel time between a pair of origins and destinations based on a comprehensive profile of historical trajectories. *Given a travel time query* $[o, d, t]$*, the goal of TTE is to predict the travel time from origin $o$ to destination $d$ utilizing historical trip and external attributes such as departure time, average speed, and so on.*

**Setup:** Before comparing the utility of two datasets for travel time estimation tasks, we first project the trajectory onto the road network using map-matching methods [36], which is well-established in relevant studies [33, 38]. Thus, the trip trajectory can be represented as a sequence of road segment information, which includes the number of paths covered by the trajectory, and the average speed on each path, among other details. We adopt four representative travel time estimators covering machine learning and state-of-the-art deep learning methods, including TEMP [30], XGBoost [7], WDR [33], and DeepTTE [11]. For experimental integrity, we divide both the original and synthetic datasets into training, validation and test sets in a 7:1:2 ratio for cross-validation. Detailed descriptions and implementations of these methods can be found in **Appendix** B.

**Result analysis:** Table 4 illustrates the performance of the original and synthetic datasets in the context of travel time estimation. Across all metrics, the synthetic dataset exhibits similar performance to the original dataset, highlighting its viability for TTE task. The difference ratios are generally low, with the highest observed difference being 10.87% in the RMSE metric for the DeepTTE model. This suggests that the synthetic dataset maintains a high degree of similarity to the original dataset, even when complex models is applied. Notably, DeepTTE shows the highest difference ratio among all models, especially in RMSE and MAE. This may indicate that more sophisticated deep learning methods, tend to be more sensitive to complex relationships and potential noise in the data. As a result, synthetic data, despite its high fidelity, may not fully recapitulate some subtle features present in the original dataset. This difference may result in the DeepTTE model appearing overfitted on the dataset and thus showing a large variation. Nevertheless, the performance on the synthetic dataset remains robust and competitive. In general, these results demonstrate the utility and validity of our synthetic datasets in the field of travel time estimation. They support the premise that such synthetic datasets can actively contribute to the progress of urban traffic analysis research.

# 6 Limitations and Future Works

While our work represents a significant stride in synthesizing GPS trajectory data for urban traffic mobility analysis, it is not without limitations. Firstly, the generation process necessitates raw data as the foundation for synthesis, meaning it cannot be generated from scratch. Consequently, the quality and characteristics of the raw data can influence the synthesized data. Secondly, the generation process is computationally demanding, requiring substantial computational resources. However, it is worth noting that the computational cost is still less than the expenses associated with actual data collection, rendering it a practical option for many researchers. In future work, we plan to further refine our trajectory synthesizer to mitigate these limitations. We aim to explore methods for reducing the computational cost of data generation and investigate ways to enhance the quality and utility of synthetic data. We are confident that our work lays a solid groundwork for future research in this area and holds the potential to significantly advance the field of urban mobility analysis.

# 7 Conclusion

This paper introduces a high-fidelity synthetic trajectory dataset for urban mobility analysis, capable of generating data with spatial-temporal properties highly consistent with real-world datasets. It addresses the prevalent limitations related to data availability, privacy issues, and data quality in existing GPS trajectory datasets. We provide a detailed and comprehensive description of this dataset and rigorously analyze it in comparison to its real-world counterpart. The validation results attest to the high-fidelity of the synthetic dataset and its substantial potential for urban traffic analysis. Furthermore, use cases indicate that synthetic trajectory datasets hold great promise for specific applications in urban mobility analysis, such as traffic demand prediction and travel time estimation. By providing a reliable source of synthetic trajectory data, we hope to facilitate further research and innovation in these areas. In conclusion, the synthetic trajectory dataset presented in this paper constitutes a significant contribution to the field of urban mobility analysis. We encourage researchers and practitioners to utilize it in their work and eagerly anticipate the advancements it will catalyze.

# 8 Acknowledgments

This research was partially supported by Research Impact Fund (No. R1015-23), APRC - CityU New Research Initiatives (No.9610565, Start-up Grant for New Faculty of City University of Hong Kong), CityU - HKIDS Early Career Research Grant (No.9360163), Hong Kong ITC Innovation and Technology Fund Midstream Research Programme for Universities Project (No.ITS/034/22MS), Hong Kong Environmental and Conservation Fund (No. 88/2022), SIRG - CityU Strategic Interdisciplinary Research Grant (No.7020046, No.7020074), Tencent (CCF-Tencent Open Fund, Tencent Rhino-Bird Focused Research Fund), Huawei (Huawei Innovation Research Program), Ant Group (CCF-Ant Research Fund, Ant Group Research Fund) and Kuaishou.

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

# A  Synthetic Dataset Analysis for Syn-Xi'an

In this section, we will perform a detailed analysis of the SYN-XI'AN dataset. By analyzing the trajectory datasets synthesized from totally different cities, we can further verify the capability of the trajectory synthesizer and the feasibility of the trajectory synthesis solution.

## A.1  Dataset description

As presented in Table 5, the SYN-XI'AN dataset possessed the same data format as the SYN-CHENGDU dataset. Specifically, the dataset also consists of one million trajectory records, each of which is represented in two parts: attribute information and trip trajectory. For these attribute information, such as `departure time` $t_d$, `trip distance (meters)`, `trip time` $t_a$`(seconds)`, and `sampling points` $n$, the representation and calculation are the same as for the SYN-CHENGDU. This consistent representation of data can save researchers substantial time, and can also support cross-city transfer learning studies.

Table 5: Dataset description of SYN-XI'AN

| Type | Description |
| --- | --- |
| Format | pickle / geoparquet |
| Size | 4.66 GB |
| Value type | float64 |
| Time frame | 5 min |
| Sample interval | 3 s |
| Spatial coverage | lat:  $34.20° \sim 34.28°$
lng: $108.90° \sim 108.99°$ |

## A.2  Trajectory geo-distribution insight

Figure 5 shows a trajectory visualization of the synthetic dataset along with a comparison of the original counterpart. Among them, the comparison between the original trajectory (Figure 5(a) ) and the generated trajectory (Figure 5(b) ) clearly shows that the synthetic dataset can well mirror the trajectory distribution of the original data. Further area zooming results show that there is no remarkable variation between the synthetic trajectory and the urban road network. Finally, the figure 5(c) exhibits the diversity of synthetic trajectories, where trajectories with the same start and end areas cover different paths.

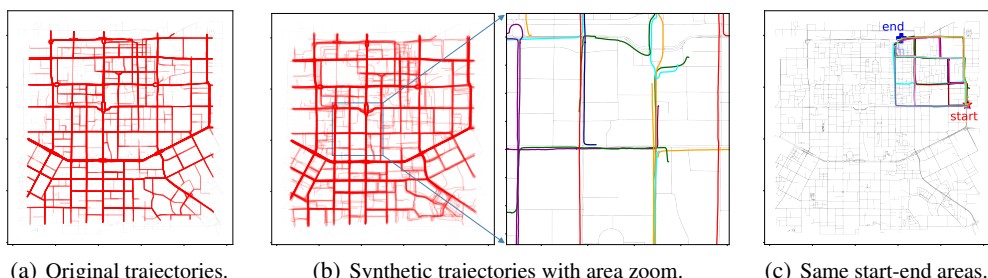

(a) Original trajectories.     (b) Synthetic trajectories with area zoom.     (c) Same start-end areas.

Figure 5:  Comparative visualizations of original and synthetic trajectories. (a) and (b) server as a reference for the quality and accuracy of the synthetic trajectories. (c) depicts the diversity of synthetic trajectories based on the same start and end areas.

## A.3  Spatial-temporal distribution

We further analyze the spatial and temporal distributions incorporated in the SYN-XI'AN dataset. As shown in the heatmap of the trajectory distribution plotted by 6(a), the synthetic trajectory dataset maintains the original distribution properties well in terms of spatial distribution. For the temporal distribution of this synthetic dataset, it is clear observe that the variation in speed and trip volume

is consistent with the real world (see Figure 6(b) and Figure 6(c)). These changes in urban traffic activity levels are not only reflected in Xi'an the city, but are also similar to the results presented in the Chengdu dataset, providing confidence in the cross-city urban mobility analysis.

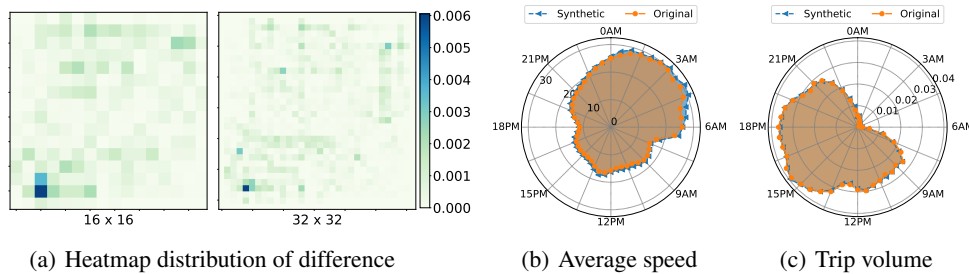

(a) Heatmap distribution of difference      (b) Average speed      (c) Trip volume

Figure 6: Spatial-temporal distribution of the synthetic dataset (SYN-XI'AN). (a) Differences in heatmap of raw-synthetic (calculated by absolute) trajectories distribution. (b) and (c) The average speed and the number of trips throughout the day.

## A.4 Trajectory properties

For the analysis at the trajectory properties level, we follow the same way as introduced in Section 4.2. The results are shown in Figure 7, where the K-S statistical similarities for the four aspects of travel distance, travel time, speed and relative distance are $0.99$, $0.94$, $0.94$ and $0.98$, respectively. In addition, the visualization results can adequately demonstrate the ability of the synthetic dataset to maintain the statistical characteristics of realistic trajectories with high fidelity. This excellent result is attributed to two aspects, the superior spatial-temporal generation performance of the proposed trajectory synthesizer. Second, the high quality of the original data further enhances the robustness of the synthesized data.

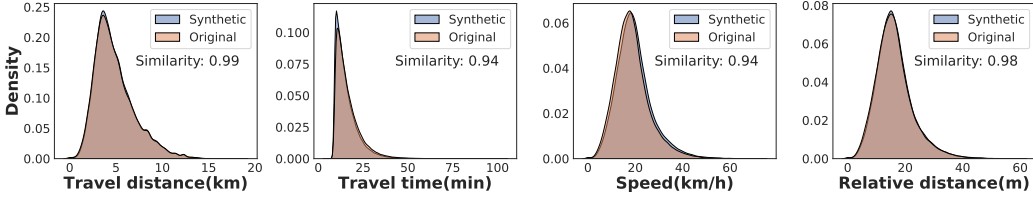

Figure 7: Comparative analysis of trajectory properties in original and SYN-XI'AN datasets.

In summary, we can conclude the following from the above analysis:

- The analysis results on the synthetic dataset of Xi'an city fully validate the feasibility of the proposed trajectory synthesis approach. It breaks a new direction for future urban mobility analytics, solving the problem of restricted use of trajectory data.
- Analytical results from two synthetic datasets demonstrate the powerful ability to employ a diffusion model as a trajectory synthesizer. This provides a novel solution for trajectory synthesis with generative models.
- Since both datasets use the same format and data style, this offers the required dataset for cross-city urban mobility analysis.

# B  Use Cases of Synthetic Dataset for Syn-Xi'an

In this section, we also use two case studies to test the utility of SYN-XI'AN. All the code was implemented by pytorch and run on a server with an Nvidia 2080 Ti GPU and Intel Silver CPU. All codes for these models follow the public benchmark[10] shown in the literature [16], and the results were averaged over 3 times. For performance evaluation of these tasks and models, we used the following metrics:

$$\text{RMSE} = \sqrt{\frac{1}{n} \sum_{i}^{n} (y_i - \hat{y}_i)^2}, \tag{1}$$

$$\text{MAE} = \frac{1}{n} \sum_{i}^{n} |y_i - \hat{y}_i|, \tag{2}$$

$$\text{MAPE} = \frac{1}{n} \sum_{i}^{n} \left| \frac{y_i - \hat{y}_i}{y_i} \right|, \tag{3}$$

where $y_i$ and $\hat{y}_i$ are the ground truth and predicted traffic value, respectively.

## B.1  Traffic Demand Prediction

In the experimental setup, we used the following advanced spatial-temporal neural network models:

- **AGCRN:** This is an adaptive convolutional recurrent neural network which contains a Node Adaptive Parameter Learning (NAPL) module and a Data Adaptive Graph Generation (DAGG) module. Where NAPL is used to capture node-specific patterns and DAGG is used to infer relationships between different traffic series.

- **GWNet:** This is a traffic prediction model based on graph convolutional network (GCN) and Wavenet structure. Among them, GCN is used to capture the spatial dependency of traffic nodes and Wavenet is used to capture the temporal dependency.

- **DCRNN:** This is a advanced neural network model based on directed graphs, which models the change of traffic flow as a diffusion process.

- **MTGNN:** This is a generic graph neural network framework that combines external knowledge and relationships between variables through a graph learning module, and then captures spatial and temporal dependencies using mix-hop propagation layers and inflated inception.

Table 6: Data utility comparison by traffic demand prediction. Results are expressed as (original / synthetic / difference ratio).

| Methods | AGCRN | GWNet | DCRNN | MTGNN |
|---------|-------|-------|-------|-------|
| RMSE | 6.51 / 6.08 / 6.61% | 6.52 / 6.39 / 1.99% | 6.52 / 6.41 / 1.69% | 6.31 / 5.97 / 5.39% |
| MAE | 4.47 / 4.16 / 6.94% | 4.47 / 4.38 / 2.01% | 4.49 / 4.37 / 2.67% | 4.41 / 4.11 / 6.80% |
| MAPE | 30.35 / 30.55 / 0.66% | 30.82 / 32.53 / 5.55% | 30.83 / 32.65 / 5.90% | 29.19 / 29.94 / 2.57% |

As presented in Table 6, we extend our evaluation to a different synthetic dataset by gauging the performance of AGCRN, GWNet, DCRNN, and MTGNN models on traffic demand prediction tasks. The metrics employed for performance evaluation include RMSE, MAE, and MAPE. Overall, the performances of these models on both the original and new synthetic datasets are close, endorsing the ability of synthetic datasets to replicate real-world scenarios in traffic demand prediction. To be specific, the difference ratio across performance metrics typically falls within a modest range, highlighting the robustness of our synthetic dataset. It is worth observing that the MAPE for GWNet and DCRNN show a slightly increased difference ratio. Despite these model-specific variances, the synthetic dataset continues to exhibit promising utility. In summary, these findings confirm the usability of our SYN-XI'AN dataset for traffic demand prediction tasks, and also expand its potential applicability in different urban traffic scenarios.

---

[10]https://github.com/deepkashiwa20/DL-Traff-Graph

## B.2 Travel Time Estiamtion

For the travel time estimation task, we used the following methods to evaluate the utility of the dataset:

- **TEMP:** The method counts trips with the same or nearby origin and destination areas and then estimates the travel time by averaging all related trips.
- **XGBoost:** The method takes travel information (e.g., distance, departure time, etc.) for each trip as input, and then estimates the travel time using an ensemble learning approach.
- **WDR:** This is a popular travel time estimation method, which estimate travel time through a combination of wide network, depth network, and recurrent network.
- **DepTTE:** This is one of the representative travel time estimation methods. It first converts the original GPS trajectory into a series of high-dimensional features, and then applies RNN to capture the spatial-temporal dependence.

Table 7: Data utility comparison by travel time estimation. Results are expressed as (original / synthetic / difference ratio)

| Methods | TEMP | XGBoost | WDR | DeepTTE |
|---------|------|---------|-----|---------|
| RMSE | 345.35 / 325.44 / 5.77% | 282.11 / 266.57 / 5.51% | 221.90 / 244.16 / 10.03% | 169.26 / 172.51 / 1.92% |
| MAE | 230.34 / 215.86 / 6.29% | 190.42 / 179.46 / 5.76% | 130.51 / 137.82 / 5.60% | 108.25 / 112.71 / 4.12% |
| MAPE | 22.21 / 21.14 / 4.82% | 18.95 / 18.09 / 4.54% | 12.12 / 12.60 / 3.96% | 10.94 / 11.10 / 1.46% |

Table 7 showcases the performance of the synthetic dataset in the context of travel time estimation for Xi'an city. Again, TEMP, XGBoost, WDR, and DeepTTE are employed for comparison, and the results are evaluated based on RMSE, MAE, and MAPE metrics. Although the absolute performance figures change due to the distinct characteristics of the new city data, the consistency between the original and synthetic datasets remains robust across the metrics and models. The difference ratio remains in a similar range, indicating flexibility and reliability of the SYN-XI'AN dataset when applied to different cities. Noteworthy is the reversed performance difference in RMSE for the WDR model, showing a $10.03\%$ discrepancy. This divergence can be attributed to the unique spatial-temporal characteristics of the different city data, indicating the need for context-specific fine-tuning when transferring models between cities. Nonetheless, the synthetic dataset still provides a reasonable approximation for most models and metrics, demonstrating its generalizability across different urban environments. In summary, these findings further reinforce the utility of our synthetic dataset, establishing its value not only for travel time estimation but also for the broader scope of urban mobility research in various cities.

# C  Additional Visualization for Use Cases

This section shows the visualizations of use case results (Section 5 and Appendix B).

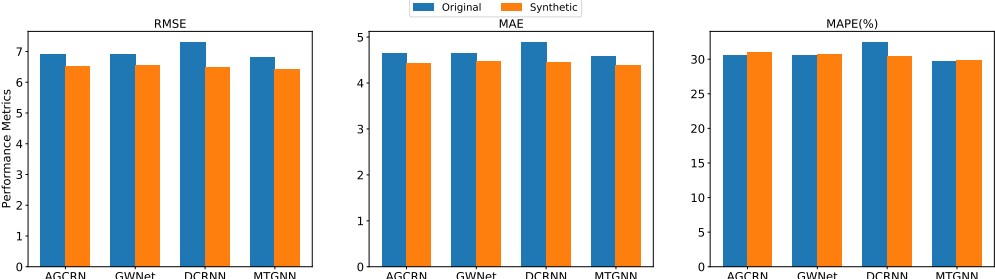

Figure 8: Performance metrics visualization for traffic demand prediction (Chengdu).

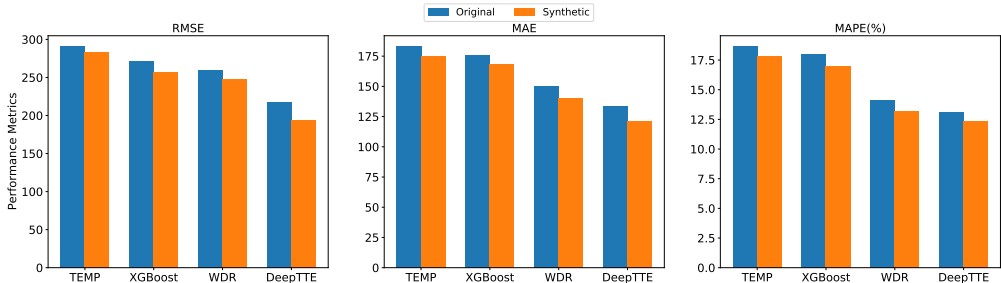

Figure 9: Performance metrics visualization for travel time estimation (Chengdu).

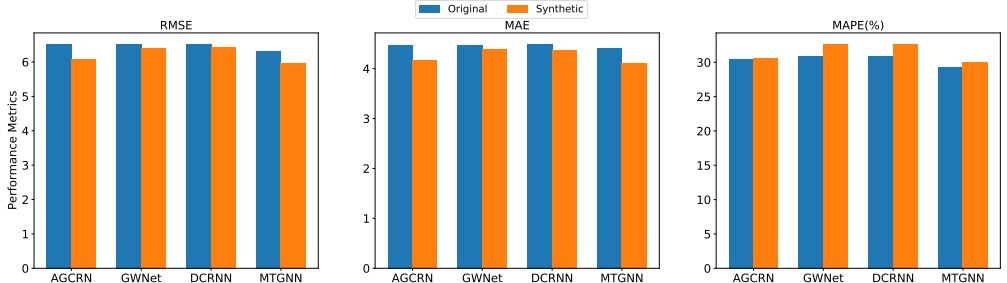

Figure 10: Performance metrics visualization for traffic demand prediction (Xi'an).

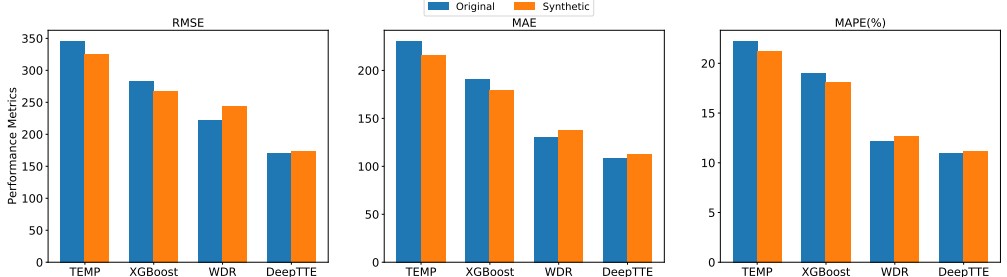

Figure 11: Performance metrics visualization for travel time estimation (Xi'an).

