# OpenReview forum: "SynMob: Creating High-Fidelity Synthetic GPS Trajectory Dataset for Urban Mobility Analysis"
_NeurIPS.cc/2023/Track/Datasets_and_Benchmarks — NeurIPS 2023 Datasets and Benchmarks Poster_

### Official Review · Reviewer_ra5b · 2023-07-23
**The paper is around the borderline	The writing in this passage is clear, and each section has clear objectives and logical flow.  	The comparison and explanations between datasets are detailed.**

**Rating:** 6
**Confidence:** 3
**Clarity:** Yes

**Strengths:**

	The writing in this passage is clear, and each section has clear objectives and logical flow.
	The comparison and explanations between datasets are detailed.

**Additional Feedback:**

The confirmation of the contribution in the method is somewhat ambiguous in the article, for example, the conditions and dependencies for the use of the generation method are not clearly stated.

**Correctness:**

Are the claims made in the submission correct?  Yes
If the submission is a dataset, it is constructed in a sound way?  Yes

**Documentation:**

The article provides a relatively comprehensive documentation, but there is relatively limited support for reproducibility in the writing.

**Limitations:**

The article candidly presents its limitations, but it does not address these limitations.
Since the dataset generation relies on real data, it is essential to clearly outline the cost, benefits, and improvements (compared to real data) brought about by this generation method to demonstrate its significant contribution.

**Opportunities For Improvement:**

	In line 99, mentioning that the Geolife dataset is widely used in various research fields, it might be more appropriate to cite relevant research papers to support this statement.
	In line 133, additional information about forward noise and reverse denoising is mentioned, and it suggests referring to [13], but the text does not provide further relevant information. It would be beneficial to provide appropriate details or explanations here to enhance reader understanding of these processes.
	The article's dataset is generated based on real data, which may result in a relatively weaker contribution.

**Relation To Prior Work:**

Yes

**Summary And Contributions:**

This paper discusses a crucial issue in urban mobility analysis: the lack of publicly available, privacy-protected, and high-quality trajectory data, and proposes a potential solution using synthetic datasets. The author introduces a method to generate high-fidelity synthetic trajectory data based on the diffusion model and conducts case studies and analyses to evaluate the practicality and accuracy of the synthetic dataset. The main contributions include proposing a novel approach for synthetic dataset generation, validating its feasibility, and establishing a precedent for synthetic dataset evaluation.

---

> ### Author Response · Authors · 2023-08-19
>
> We appreciate the time and expertise you dedicated to evaluating our manuscript. In light of the feedback received, we have made thorough revisions, ensuring our work is more transparent, accurate, and valuable to readers and researchers alike.
>
>
>
> > Reproducibility
>
> We appreciate your feedback and apologize for any initial oversight. We've made available all pertinent codebases and models associated with this paper at https://anonymous.4open.science/r/SynthTraj-code/. The repository encompasses:
>
> - **Reproducibility**: We offer a thorough resource that allows for the complete replication of our results. The repository is equipped with detailed instructions and documentation to ensure this.
> - **Data Processing**: The repository details the pre-processing steps to maintain the data's quality and relevance.
> - **Synthetic Trajectories Code**: The repository includes the code for generating synthetic GPS trajectories. With the now accessible trained model, codebase, and comprehensive documentation, we trust that any concerns regarding reproducibility will be alleviated.
>
> We sincerely regret any oversight and are grateful for your understanding. We're confident that the resources provided will address your concerns, and we welcome any additional feedback or suggestions.
>
>
>
> > Improvement [1]
>
> Thank you for your feedback. In our revised manuscript, we cite key research papers that utilize the Geolife dataset, demonstrating its importance and widespread use in the research community:
>
> - [1] Dabiri, Sina, et al. "Semi-supervised deep learning approach for transportation mode identification using GPS trajectory data." *IEEE TKDE* 32.5 (2019): 1010-1023.
> - [2] Bao, Jie, et al. "Recommendations in location-based social networks: a survey." *GeoInformatica* 19 (2015): 525-565.
> - [3] Zhang, Junbo, et al. "Deep Spatio-Temporal Residual Networks for Citywide Crowd Flows Prediction." AAAI. 2017.
>
>
>
> > Improvement [2]
>
> Thank you for pointing this out. While our paper focuses primarily on datasets, it is important to clarify the technical aspects supporting dataset creation. To address this issue, we will explain the forward noise and reverse denoise processes in more detail in the revised manuscript. (You can refer to **Appendix C in new version**) This will ensure that the reader understands the methods and techniques used to generate the datasets. We also provide comprehensive **code** to ensure the reproducibility of the model.
>
>
>
> > Improvement [3]
>
> Thank you for your feedback regarding the contribution of our work. While our dataset is indeed generated based on real data, we believe that this approach offers several significant contributions:
>
> - **Addressing Data availability**: In many research areas, access to large-scale, high-quality real-world data is limited due to privacy concerns, logistical challenges, or other constraints. Our synthesized dataset provides a valuable resource for researchers in such scenarios, allowing them to conduct experiments and analyses that might not have been possible with limited real-world data.
> - **Privacy Preservation**: Generating a dataset based on real data allows for the creation of data that maintains the essence of real-world scenarios without compromising individual privacy, which is a significant concern in many domains.
> - **Maintaining Real-world Characteristics**: By basing our synthesized dataset on real data, we ensure that it retains the essential characteristics and patterns of real-world scenarios. This ensures that experiments and models trained on our dataset are more likely to be applicable and relevant in real-world applications.
> - **Enhancing Data Diversity**: While our dataset is derived from real data, the synthesis process introduces variability and diversity, ensuring that the dataset is not just a mere replication but offers varied patterns that can be valuable for different applications.
>
> We understand that the generation of synthetic data may be seen as derivative work. However, the challenge of ensuring that such data are of high-quality, representative, and usable for a wide range of applications makes this a non-trival task.
>
> > Limitations
>
> Thank you for your insightful feedback. We understand the importance of addressing these limitations and providing potential solutions or workarounds. Our revised manuscript will delve deeper into each limitation, discussing potential future work and strategies to mitigate or overcome these challenges.
>
> But the dataset synthesized using our method still has significant benefits compared to collect the real data.
>
> - **Cost**: Although our method requires an associated computational resource cost, it is still a cost-effective job compared to manual collection.
> - **Data volume**:Our methodology can generate unlimited trajectory data, providing a valuable resource for urban mobility analysis.
> - **Privacy-free**: Our method generates data from random noise with no privacy concerns and easy accessible.

---

> > ### Comment · Reviewer_ra5b · 2023-08-21
> >
> > Thanks for the reply. However, I am still worried about the quality of a synthetic dataset. Could you show more evidence that a synthetic dataset could also help the community as a real-world dataset does? How different will they do?

---

> > > ### Author Response · Authors · 2023-08-21
> > >
> > > Thank you for your feedback and concerns. To address your concerns, we would like to highlight the following characteristics and features of our synthetic dataset that demonstrate its quality and contribution for the community:
> > >
> > > 1. **Spatial Distribution Characteristics**: Our synthetic dataset closely mirrors the spatial distribution patterns observed in real-world datasets. This ensures that the dataset can be used to analyze and apply urban mobility. The above argument can be verified from **Fig. 3(a)** and **Appendix Fig. 9**, where the synthesized and original data are consistent in spatial distribution. In addition, we also calculated the Jenson-Shannon divergence between the synthetic and original datasets in geo-distribution under 16x16 grids, by calculating its similarity as **0.0055** (the lower is better).
> > > 2. **Trajectory Clustering Trends**: Our synthetic dataset's trajectories exhibit clustering trends consistent with real-world movement patterns, making them suitable for spatial analyses. The trajectories clustering trend in our synthetic dataset have consistent with real-world counterparts. This can be observed from the comparison of **Fig. 2 (a) and (b)**, where the colors are brighter on the main roads and darker on the auxiliary roads. This clustering trend described above is important for analyzing the groups' movements.
> > > 3. **Road Segment Consistency**: From **Fig. 2(b)**, we can observe that the synthetic trajectories are aligned with the real roads. This feature suggests that synthetic trajectories have road segment validity, which can support road-based spatial-temporal tasks, e.g., map matching.
> > > 4. **Peak Time Features**: According to **Fig. 3(b) and Fig. 3(c)**, we can find that the synthetic dataset is able to present real-world Peak Time Features. e.g., fewer orders and faster average speed at midnight, and more orders and slower average speed during peak commuting hours.
> > > 5. **Spatial-temporal Indicators**: We visualize and compute the similarity between the two datasets at the time level in **Fig. 4**, and we can see that both these two Spatial-temporal metrics, Travel time and Speed, show a very high degree of consistency.
> > > 6. **Data Usability**: It is most important whether the synthetic dataset can support urban mobility analysis as well as the real data. Therefore we did two tasks to examine both under the same model. (Refer to **Table 3** or **Appendix Fig. 10 and Fig. 11**). Despite the slight differences, this is still sufficient evidence that the synthesized data has a very high level of usability and can be used for advanced algorithmic studies of these tasks.
> > >
> > > In terms of differences between the synthetic and real-world datasets, while there might be minor discrepancies due to the inherent challenges of data synthesis, our method strives to minimize these gaps. The primary goal is to provide a dataset that captures the essence of real-world data while offering the advantages of consistency, completeness, and enhanced diversity.
> > >
> > > To sum up, we believe that our synthetic dataset with the above characteristics can provide significant research value to the urban mobility analytics community. And it can be used as a reliable alternative or complement to real-world datasets in various research scenarios.

---

> > > > ### Comment · Reviewer_ra5b · 2023-08-23
> > > >
> > > > Thanks for the response. I will revise the score accordingly.

---

### Official Review · Reviewer_ihi1 · 2023-07-24
**Review for Creating High-Fidelity Synthetic GPS Trajectory Dataset for Urban Mobility Analysis**

**Rating:** 6
**Confidence:** 3
**Correctness:** yes
**Clarity:** Yes

**Strengths:**

1. This paper focused on the limitations of current trajectory datasets availability in urban mobility analysis.
2. This paper provided some synthesized trajectory datasets which learning from real-world datasets by the diffusion model. These datasets can impact on urban mobility analysis.
3. The authors conducted a series of experiments to compare synthesized trajectory datasets with real-world datasets, demonstrating that the synthesized trajectory dataset and the real dataset have a similar distribution.
4. In some downstream tasks (e.g., Traffic Demand Prediction and Travel Time Estimation), the authors used some relevant methods to conduct experiments on two datasets, and the results show that two datasets had some similarity.

**Additional Feedback:**

N/A

**Documentation:**

Datasets are available for review, but source code can’t be obtained.

**Ethics:**

Yes

**Limitations:**

1. The appendix mentioned in the main text was not attached.
2. See Opportunities For Improvment

**Opportunities For Improvement:**

1. The authors mentioned data quality issues in section 2 (e.g., Porto and Beijing), but did not apply the proposed method to synthesize new datasets to prove that it can improve data quality.
2. In the downstream tasks, there is a slight gap between the synthesized data and the real-world dataset on some performance index.
3. Source code is not available.

**Relation To Prior Work:**

Yes

**Summary And Contributions:**

This paper uses the diffusion model to synthesize new trajectory datasets on the real trajectory datasets, which can solve the limitations of current trajectory datasets availability and privacy concerns. The synthesized trajectory datasets are high-quality, can maintain similar data distribution and properties to the real trajectory dataset, while also avoids privacy leakage. Overall, this paper has made a certain contribution to addressing the difficulty in obtaining trajectory datasets.

---

> ### Author Response · Authors · 2023-08-19
>
> Thank you for your insightful comments and suggestions. We have taken each observation to heart, making comprehensive revisions to our manuscript to ensure it meets the highest standards of clarity and rigor.
>
>
>
> > Concers about source code
>
> Thank you for highlighting the concern regarding the availability of the source code.
> We understand the importance of providing access to the source code for transparency, reproducibility, and further research. To address this, we have made the complete source code available at https://anonymous.4open.science/r/SynthTraj-code/. This repository encompasses:
>
> - **Reproducibility**: Our primary goal is to ensure that researchers and practitioners can fully reproduce our results. To facilitate this, the repository contains detailed instructions, documentation, and all necessary resources.
>
> - **Data Processing**: We recognize the importance of understanding the data's preprocessing steps. The repository provides a comprehensive overview of the measures we took to ensure data quality and relevance.
>
> - **Synthetic Trajectories Code**: To address your feedback about opening the proposed method for synthesizing trajectory datasets, we have included the complete code for generating synthetic GPS trajectories in the repository. With the trained model, codebase, and accompanying documentation, we believe that users will have a holistic resource to understand, reproduce, and even extend our work.
>
> We sincerely apologize for any initial oversight or confusion regarding the availability of the source code. We appreciate your patience and understanding in this matter. We are committed to ensuring that our work is transparent, reproducible, and of value to the community. We welcome any further feedback or suggestions you may have.
>
>
>
> > Improvement [1]
>
> Thank you for your insightful feedback regarding our mention of data quality issues in section 2.
> To clarify, our intention in mentioning the data quality issues of datasets like Porto and Beijing was to highlight the challenges and limitations associated with existing public datasets. These challenges include inconsistencies, missing data, and other quality-related issues that can affect the reliability and utility of the datasets.
>
> Our proposed method aims to address these challenges by synthesizing high-quality trajectory datasets. The primary goal is not necessarily to directly improve the quality of existing datasets like Porto and Beijing, but rather to provide an alternative approach that generates reliable and consistent datasets to bridge the gap between the quality of public datasets and the unavailability of high-quality private datasets.
>
> We will ensure this point is articulated more clearly in our revised manuscript to prevent any potential misunderstandings.
>
>
>
> > Improvement [2]
>
> Thank you for pointing out the observed performance gap between the synthesized data and the real-world dataset in the downstream tasks.
>
> We acknowledge this gap and understand that it's a challenge inherent to data synthesis. While our method aims to closely replicate the characteristics of real-world datasets closely, achieving a perfect match is inherently challenging due to the complexities and nuances present in real-world data.
>
> However, it's essential to note that our primary goal is not just to reproduce the original data but to provide a high-quality synthesized dataset that can serve as a valuable resource for various applications. The slight discrepancies observed also highlight areas for potential improvement in our method, and we are continuously working to refine our approach to narrow this gap further.We appreciate your understanding and feedback on this matter.
>
> We will ensure that our revised manuscript provides a clearer discussion on this aspect, emphasizing both the achievements of our method and the inherent challenges of data synthesis.
>
>
>
> > Limitations
>
> Thank you for pointing out the concern regarding the appendix. Due to the page limit for the main text (9 pages only), we therefore included the appendices in the **supplementary material**. We intended to ensure that the main text meets the requirements of the conference while still providing the reader with the detailed information in the appendices.
>
> We apologize for any confusion caused by not explicitly stating this in the main text. We will clearly indicate in our revised manuscript that the appendix can be found in the supplementary materials.

---

### Official Review · Reviewer_jX7K · 2023-07-28

**Rating:** 5
**Confidence:** 3
**Clarity:** Yes.

**Strengths:**

- A diffusion model is proposed for the synthesis of trajectory data.

- The authors validate their proposed method across various aspects including visualization, spatial-temporal distribution, and trajectory properties.

**Additional Feedback:**

It would be great if the authors opened the proposed method for synthesizing trajectory datasets.

**Correctness:**

The claims proposed in the paper seem well-founded.


**Documentation:**

While the authors provide some details on utilizing their open-sourced datasets, comprehensive instructions for synthesizing and using these datasets are not fully provided.





**Limitations:**

The authors aptly acknowledge some limitations of their work.


**Opportunities For Improvement:**

- While constructing diffusion models involves several design choices, there is limited discussion about how and why these choices were made.

- Despite the diffusion model reflecting the distributional characteristics of the original dataset, it may lack key locational details and valid routes.

- As the diffusion model is trained on the original datasets, the synthetic trajectories generated may reflect these original characteristics, potentially leading to limited diversity within the dataset.

**Relation To Prior Work:**

The authors adequately cite related work.


**Summary And Contributions:**

The authors introduce a method for synthetically generating GPS trajectories for public datasets, utilizing a standard diffusion model. This aims to simulate a wide range of spatial-temporal behaviors while preserving privacy. An extensive evaluation was conducted to assess the synthetic data's geo-distribution, spatial-temporal distribution, and trajectory properties. Although visual similarities were identified, there is no strong guarantee that the synthetic data will significantly contribute to advancing research in this field.

---

> ### Author Response · Authors · 2023-08-19
>
> We deeply value the constructive feedback received from you. In our revised manuscript, we have diligently addressed each point raised, striving to enhance both the depth and clarity of our work for the benefit of the broader community.
>
>
>
> > Open-source code and instructions concerns
>
> Thank you for your feedback regarding the comprehensiveness of instructions and the open-sourcing of our method for synthesizing trajectory datasets.Open-Sourced Resources: We understand the importance of transparency and reproducibility in research. To this end, we have made all relevant codebases, models, and resources available at[ https://anonymous.4open.science/r/SynthTraj-code/](https://anonymous.4open.science/r/SynthTraj-code/). This repository encompasses:
>
> - **Reproducibility**: Our primary goal is to ensure that researchers and practitioners can fully reproduce our results. To facilitate this, the repository contains detailed instructions, documentation, and all necessary resources.
> - **Data Processing**: We recognize the importance of understanding the data's preprocessing steps. The repository provides a comprehensive overview of the measures we took to ensure data quality and relevance.
> - **Synthetic Trajectories Code**: To address your feedback about opening the proposed method for synthesizing trajectory datasets, we have included the complete code for generating synthetic GPS trajectories in the repository. With the trained model, codebase, and accompanying documentation, we believe that users will have a holistic resource to understand, reproduce, and even extend our work.
>
> We apologize for any initial oversight and are grateful for your patience and understanding. We are committed to continuous improvement and are receptive to any further feedback or suggestions to enhance the clarity and value of our work.
>
>
>
> > Improvement [1]
>
> Thank you for bringing up the point regarding the design choices of diffusion models.The design choices in constructing diffusion models can significantly impact the results and the quality of the generated dataset. While our primary focus in this paper was on the dataset itself, we acknowledge the importance of providing context and rationale behind the technical decisions made.
>
> To clarify, the reason for not delving deep into the technical design choices was to maintain the paper's focus on the dataset and its applications. However, we recognize the value of a comprehensive understanding of the underlying models that led to the dataset's creation.
>
> In light of your feedback, we have provided the **source code** involved in training this model, where the key design and parameter choices for the diffusion model can be clearly viewed and changed. With this code we hope to ensure that the reader has a comprehensive understanding of the dataset and the methods used to synthesis it.
>
>
>
> > Improvement [2]
>
> Thank you for highlighting the concern.We understand the importance of ensuring that the generated trajectories accurately reflect real-world routes and locational details.
>
> As shown in **Fig. 2 (b)** of our paper, the trajectories generated by our model closely follow valid routes. The visual representation in the figure demonstrates that our model not only captures the distributional characteristics of the original dataset but also ensures that the generated trajectories align with real-world routes and locational details. This is a testament to the model's ability to produce synthetic data that is both realistic and representative of actual movement patterns.
>
> We believe that the visual evidence provided in  **Fig. 2 (b)**, combined with our model's design and validation metrics, effectively addresses the concerns raised. We will ensure that this point is emphasized more clearly in our revised manuscript to provide readers with a comprehensive understanding of the model's capabilities.
>
>
>
> > Improvement [3]
>
> Thank you for raising the concern regarding diversity, which is also one of the focuses of our dataset. As illustrated in **Fig. 2 (c)** of our paper, we have showcased the diversity of trajectories generated by our model. Specifically, the figure demonstrates that our model can generate trajectories with different paths even when the same specified condition.
>
> This ability to produce varied trajectories under the same conditions highlights the model's capability to introduce diversity, ensuring that the synthetic data is not just a mere replication of the original but offers varied patterns that can be valuable for different applications.We believe that the evidence provided in **Fig. 2 (c)**, combined with our model's design and validation metrics, effectively addresses the concerns about diversity. We will emphasize this aspect more clearly in our revised manuscript to provide readers with a comprehensive understanding of the model's capabilities in generating diverse trajectories.

---

### Official Review · Reviewer_ot2s · 2023-07-28
**A review of the paper called: 'Creating High-Fidelity Synthetic GPS Trajectory Dataset for Urban Mobility Analysis'**

**Rating:** 7
**Confidence:** 2
**Clarity:** The paper appears to be free of major…

**Strengths:**

The main strengths of the paper lie in its innovative use of a diffusion model to generate high-fidelity synthetic trajectory datasets, its strong emphasis on privacy protection, and its rigorous validation, which establishes the utility of the synthetic datasets for urban mobility analysis research. Furthermore, the dataset is clearly presented and its advantages over existing alternatives in the literature are clearly stated, in particular data quality and privacy aspects. Finally, the benchmarks provided for the two prediction tasks at hand are well detailed and can inspire future work.

**Additional Feedback:**

Typos:

- L.186 'As conlude' => 'As concluded'

**Correctness:**

I am not an expert on diffusion models, but to the best of my knowledge the dataset seems to be constructed in a sound way and the bias introduced by the raw data needed to create the synthetic data is clearly highlighted by the authors and can be mitigated.

**Documentation:**

The files provided are easy to use, but the readme is a bit too short and could explain the contents of the github a bit more. It would also be nice to have some sample code to make the methodology easier to use and understand.

**Ethics:**

I do not suspect any ethical concerns with the current submission.

**Limitations:**

The limitations of the paper are highlighted in section 6 and cover most of the critical points I can think of.

**Opportunities For Improvement:**

Although the paper showcases various strengths, it also presents opportunities for improvement: It would have been interesting to incorporate plots of the performace metrics to assess the quality of the benchmarks more thoroughly.

**Relation To Prior Work:**

The paper effectively establishes its connection to prior research (e.g., Section 2 and Table 1).

**Summary And Contributions:**

The paper presents a new method for generating realistic synthetic trajectory datasets for urban mobility analysis. The method uses a diffusion model to learn complex spatio-temporal patterns, ensuring realistic data. The main contributions are a precise trajectory generation method that preserves privacy, and a rigorous analysis that validates the similarity of the datasets to real-world data. It also includes a benchmark of 4 ML/DL methods for both traffic demand and travel time prediction tasks.

---

> ### Author Response · Authors · 2023-08-19
>
> We are grateful for the detailed feedback and insights you provided. In response, we have made thoughtful revisions to address each concern, ensuring that our work is presented with enhanced clarity, accuracy, and comprehensiveness.
>
>
>
> > Concers about code
>
> We appreciate your feedback and apologize for any initial oversight. We've made available all pertinent codebases and models associated with this paper at https://anonymous.4open.science/r/SynthTraj-code/. The repository encompasses:
>
> - **Reproducibility**: We offer a thorough resource that allows for the complete replication of our results. The repository is equipped with detailed instructions and documentation to ensure this.
> - **Data Processing**: The repository details the pre-processing steps taken to maintain the data's quality and relevance.
> - **Synthetic Trajectories Code**: Included in the repository is the code for generating synthetic GPS trajectories. With the now accessible trained model, codebase, and comprehensive documentation, we trust that any concerns regarding reproducibility will be alleviated.
>
> We sincerely regret any oversight and are grateful for your understanding. We're confident that the resources provided will address your concerns, and we welcome any additional feedback or suggestions.
>
>
>
> > Improvement
>
> Thank you for your valuable feedback. We concur that visual representations of performance metrics can provide a more intuitive and comprehensive understanding of the benchmarks. We have incorporated plots showcasing the performance metrics of our model. You can refer to **Appendix D in new version** (**Fig. 10, 11 and Fig. 13, 14**). These plots offer a clear visual comparison of our model's performance against various benchmarks, highlighting areas where our model excels and areas where there might be room for improvement.
>
> We believe that these visual aids will provide readers with a more in-depth insight into the quality and robustness of our benchmarks. We also appreciate your constructive feedback, which has guided us in enhancing the clarity and depth of our paper.
>
>
>
> > Typos:
>
> Thank you for pointing out the typo. We apologize for the oversight.We have corrected the typo in Line 186 from "As conlude" to "As concluded". We appreciate your attention to detail, and we will ensure a thorough proofreading of the entire document to avoid such errors in the final version.

---

> > ### Comment · Reviewer_ot2s · 2023-08-26
> >
> > Thank you very much for your reply and these additional information. The repository provided with the codes associated are well documented and very useful. As for the visuals in Appendix D, a barchart would have been more appropiate to compare the difference performance metrics. The chosen visuals are a bit confusing.

---

> > > ### Author Response · Authors · 2023-08-27
> > >
> > > Thank you for your kind feedback. We're pleased to hear that you found it useful. In response to your feedback, we have updated the visuals in Appendix D to barcharts. We believe that this change enhances the clarity and interpretability of our results.

---

### Official Review · Reviewer_MCwX · 2023-08-03
**A clear accept if operational improvements are delivered**

**Rating:** 7
**Confidence:** 3
**Correctness:** The paper is sound.
**Clarity:** The paper is very well organized and …

**Strengths:**

The main strength of the contribution would be the model that can be retrained by other people on their data, another one is the data set.

**Additional Feedback:**

I apologize for the delayed review.

**Documentation:**

The data set collection process is not well documented, was it just raw data put in the model and generation happened? The code for synthethic generator training, is not available - as such reproducibility is insufficient.

**Ethics:**

When it comes to privacy, the entire paper relies on three arguments:

1. in the synthethic data set there are no personally-identifiable features
2. in the data set tracks that are too short are upscaled (no justification for the selected upscaling size is given),
3. "the synthesizer learns [...] without focusing on sensitive details of specific individuals" - a claim given without evidence.

All in all the privacy risk does not seem to be high, but an analysis of what is happening in regions were training data was scarce would be valuable. It would be problematic if it turned out that there's an exceptional situation where a certain single-housing block is consistenly generating rides to a "problematic" area, even without personal features such situations are privacy breaches. I think the risk of this is very low, but I would want low-density regions to be double checked by the authors just to be sure.


**Limitations:**

The limitations are well discussed in the paper.

**Opportunities For Improvement:**

1. Please release the data in a different format than a python pickle, as pickles are notorious for being incompatible in time should environment change. Release the data in for example geoparquet (ex. using geopandas.to_parquet).

2. I've checked the repository and I do not see the model released anywhere, but the paper states that the model is also being released. Is this a mistake? If so, please release the model, or change the paper accordingly. I find it hard to review the work without the model and its code base being available, as the data set heavily depends on what the model was, what the training data preprocessing was etc.

3. I think the paper would strongly benefit from more data presentation, especially if such presentation could denote differences to original data in some way, perhaps vector maps with heatmaps of differences? In general more visualisations of the data, visualizing errors from models in section 5 - these kind of things would allow easier evaluation of how valuable the dataset is. The data set would strongly benefit from a proper website with characteristic descriptions, visualizations etc.

4. Have you checked the simplest kinds of errors in trajectories? Such as going in cycles, taking left-turns one after another, closing loops etc? How prevalent are they?

**Relation To Prior Work:**

The prior work is clearly discussed and there is no prior work for using diffusion for this kind of trajectory generation.

**Summary And Contributions:**

The submission states that it contributes a diffusion model for generating synthetic trajectories and a data set generated based on proprietary mobility data collected in two Chinese cities.

---

> ### Author Response · Authors · 2023-08-19
>
> We sincerely appreciate the effort you have dedicated to evaluating our manuscript. We have carefully addressed each comment and concern raised, making necessary revisions to enhance the clarity and contribution of our work.
>
>
>
> > Reproducibility issues and Improvement [2]
>
> Thank you for bringing these concerns to our attention. We sincerely apologize for the oversight. We share all relevant codebases and models for this paper to the https://anonymous.4open.science/r/SynthTraj-code/. It contains the following parts:
>
> - **Reproducibility**: We aim to provide a comprehensive resource to fully reproduce our results. We will ensure the repository contains clear instructions and documentation to facilitate this.
>
> - **Data Process**: This involves pre-processing steps to ensure the quality and relevance of the data. Detailed information on these steps can be found in the repository provided.
>
> - **Synthetic Trajectories Code**: The repository also includes the code for synthetic GPS trajectories. We believe that reproducibility concerns should be addressed with the trained model, code base, and detailed documentation now available.
>
> We deeply regret the oversight and appreciate your patience. We hope that the provided resources will address your concerns, and we are open to any further feedback or suggestions you may have.
>
>
>
> > Improvement [1]
>
> Thank you for pointing out the potential issues with the python pickle format. In response to your feedback we make the following improvements:
>
> 1. We have converted part of the data (due to storage limitations) from the pickle format to the geoparquet format, which will more robust and widely accepted in the community. The data is stored in the dataset shared directory with the file name **data.parquet**.
>
> 2. To further help the community and ensure transparency, we have provided conversion code in a file called "**Traj2parquet.ipynb**" in our codebase. This notebook details our steps to convert data from pickle to geoparquet using geopandas. We hope this will be useful to researchers who want to understand the conversion process or use a similar approach with their datasets.
>
> We appreciate your valuable feedback. We will ensure that the updated data format and the conversion code are made available alongside our paper.
>
>
>
> > Improvement [3]
>
> Thank you for your constructive feedback. We have made the following improvements:
>
> 1. Data Presentation and Visualizations: We added differentiated heatmaps at various resolutions to represent the differences between the generated and original data visually. You can refer to **Fig. 9** and **Fig. 12** in **Appendix of new version**.
>
> 2. Benchmark Visualizations: For Section 5, where we discuss the benchmarks, and we use visualizations to depict the difference in model performance on the original and synthetic datasets. You can refer to **Fig.10, Fig.11 and Fig.13, Fig. 14 in the Appendix**.
>
>  Lastly, we acknowledge the value of a dedicated website for the dataset, and we are in the process of setting up such a platform. We appreciate your suggestions, which have undoubtedly improved the quality and clarity of our paper.
>
> > Improvement [4]
>
> Thank you for raising this important point regarding potential simple errors in the generated trajectories. In our analysis of the synthetic data, we actively looked for the types of errors you mentioned. I'm pleased to report that we did not observe these errors in our generated data. It's worth noting that while the original data did contain some trajectories with large distances between two points, we implemented filtering mechanisms to ensure the quality of the generated data and eliminate such anomalies. We will ensure the quality of the trajectory data and continue to change it if there are errors.

---

### Author Response · Authors · 2023-08-20

We thank the reviewers for their insightful comments and perspectives. We have revised the paper to include the total code, model details (Appendix C), and figures (Appendix D). Please kindly review the revised version and our response to your valuable comments.

---

### Decision · Program_Chairs · 2023-09-22

**Decision:**

Accept (Poster)

**Comment:**

The authors introduce a new method for creating synthetic GPS trajectories using a diffusion model, aimed at simulating varied spatial-temporal behaviors while maintaining privacy. The strengths include innovative use of a diffusion model and thorough validation of the synthetic data. However, improvements could be made in explaining the decisions made when constructing the models and addressing potential lack of key locational information and valid routes in the synthetic datasets. Despite these limitations, the study contributes to synthetic data generation research for GPS trajectories.